# Research on Fuzzy Control of Methanol Distillation Based on SHAP (SHapley Additive exPlanations) Interpretability and Generative Artificial Intelligence

**DOI:** 10.3390/s25051308

**Published:** 2025-02-21

**Authors:** Yuhan Gong, Qinyu Zhang, Yuxian Ren, Zhike Liu, Mohamad Tarmizi Abu Seman

**Affiliations:** 1College of Electrical Engineering, North China University of Science and Technology, Tangshan 063210, China; gongyuhan@student.usm.my (Y.G.); liuzhike@stu.ncst.edu.cn (Z.L.); 2School of Electrical and Electronic Engineering, Universiti Sains Malaysia, Nibong Tebal 14300, Penang, Malaysia; 3College of Science, North China University of Science and Technology, Tangshan 063210, China; zhangqinyu@stu.ncst.edu.cn; 4College of ‘Yisheng’, North China University of Science and Technology, Tangshan 063210, China; ryx@stu.ncst.edu.cn

**Keywords:** methanol rectification, generative artificial intelligence, type II fuzzy neural network, inverse steady-state prediction, SHAP analysis

## Abstract

The most important control parameters in the methanol distillation process, which are directly related to product quality and yield, are the temperature, pressure and water content of the finished product at the top of the column. In order to adapt to the development trend of modern industrial technology to be more accurate, faster and more stable, the fusion of multi-sensor data puts forward higher requirements. Traditional control methods, such as PID control and fuzzy control, have the disadvantages of low heterogeneous data processing capability, poor response speed and low control accuracy when dealing with complex industrial process detection and control. For the control of tower top temperature and pressure in the methanol distillation industry, this study innovatively combines generative artificial intelligence and a type II fuzzy neural network, using a GAN for data preprocessing and a type II fuzzy neural network for steady-state inverse prediction to construct the GAN-T2FNN temperature and pressure control model for an atmospheric pressure tower. Comparison experiments with other neural network models and traditional PID control models show that the GAN-T2FNN model has a better performance in terms of prediction accuracy and fitting effect, with a minimum MAE value of 0.1828, which is more robust, and an R^2^ Score of 0.9854, which is closer to 1, for the best overall model performance. Finally, the SHAP model was used to analyze the influence mechanism of various parameters on the temperature and pressure at the top of the atmospheric column, which provides a more comprehensive reference and guidance for the precise control of the methanol distillation process.

## 1. Introduction

As one of the most basic chemical raw materials, methanol is widely used in industrial production, the energy field and the chemical synthesis process, as shown in Figure 1 [1,2]. With the continuous progress of technology and the changes in energy structures, methanol not only shows remarkable strength in organic compounds, but also shows significant advantages in energy saving and emission reduction [3,4]. With fierce market competition, the increasing price of raw materials has led countries around the world to actively research efficient, energy-saving and environmentally friendly methanol production technologies to replace the overdependence on traditional petroleum and ore materials [5,6,7,8].

Coke oven gas and natural gas methanol production dominates in China’s coal methanol production, accounting for more than 97% of the total production, and the methanol tower is an important piece of synthesis equipment in the process of methanol production from coke oven gas. The temperature and pressure control effect of the methanol synthesis tower directly affects product quality and energy consumption. Since the reaction catalyzed by CO, CO_2_ and H_2_ to produce methanol is a multiphase catalytic reversible exothermic reaction [9,10], the temperature in the reformer presents nonlinearity, a significant time lag and uncertainty [11,12]. During the production process, it is crucial to control the reaction temperature strictly to maintain the catalyst activity and ensure the high productivity of methanol [13,14,15].

The PID controller is one of the most commonly used control methods in industry. With the development of industrial technology, traditional control methods have been insufficient to meet the increasingly complex control requirements. When faced with a nonlinear, time-varying and complex methanol synthesis process, the response to the dynamic changes has been slow. Because it is difficult to deal with the interactions in complex systems, sensitivity to external perturbations, model uncertainty, the lack of strong feedback and lack of a self-learning ability, the control accuracy has been low [16,17,18]. BP neural networks have the ability to learn nonlinear processes and have been widely used in industrial prediction and control [19]. Li Shengwang and Duan Xin et al. used a flexible BP (back-propagation) method in their neural network model to predict and control the opening temperature of the decomposition furnace [20], but its training speed is slow and it easily falls into local minima, which makes it difficult to meet the precise temperature control requirements of methanol synthesis [21,22,23]. The gated recurrent unit (GRU) neural network is a variant of the long short-term memory (LSTM) network, which is able to process time series data with fewer parameters and faster computational speed [24,25]. Guo Jun et al. used GRU neurons to study the linear relationship between indicator gas and temperature, and optimized the parameters of the GRU model using a particle swarm algorithm to successfully predict the temperature of the coal body [26]. However, for the complex dynamic characteristics of the methanol synthesis process, its modeling ability is insufficient, and its memory is limited.

Fuzzy control, which depends on the theory of fuzzy sets, fuzzy language variables and fuzzy logic reasoning, is better suited for handling systems with mathematical models that are difficult to obtain and have dynamic characteristics. Its main advantage is robustness, but the disadvantages are also prominent, namely, low control accuracy and poor dynamic quality [27,28]. Wenfang Li and Yuqiao Wang developed a fuzzy algorithm to modify PID parameters in real time, improving accuracy and successfully implementing intelligent temperature control of instrumentation in manufacturing processes [29]. Mario Maya et al. effectively improved the productivity of chemical reactions in the reactor by utilizing a neuro-fuzzy controller calibrated with a meta-heuristic algorithm, resulting in significantly higher biodiesel production [30]. Neural and fuzzy approaches have been successfully used in chemical process engineering processes. In recent years, the application of neuro-fuzzy systems has grown rapidly. Fernández de Cañete et al. used a fuzzy control strategy enhanced with self-learning capabilities to successfully implement the control of a binary methanol–propanol distillation column [31]. Soundar Ramchandran and R. Russell Rhinehart proposed using a neural network to represent the process’s steady-state inverse in conjunction with a reference system to create a multivariate controller for the successful dynamic simulation of two methanol–water columns for distillation [32]. In view of the high complexity, dynamic nature and volatility of the methanol production process, traditional control methods have exposed many limitations in dealing with it [33,34]. It is difficult to accurately depict the complex dynamic characteristics in the production process, and it has poor adaptability to the working conditions, with large fluctuations, resulting in unsatisfactory control effects and an inability to meet the actual production needs [35]. In this study, a type II fuzzy neural network (T2FNN), a generative adversarial network (GAN), and inverse steady-state prediction are integrated into the control system of the methanol reforming process, and a new intelligent control strategy with high efficiency and robustness is constructed. The GAN improves the data quality and compensates for the limitations of the traditional methods in small-sample and high-noise environments by augmenting and optimizing the industrial data; the T2FNN improves the adaptive capability and control accuracy of the system under complex nonlinear and time-varying environments by introducing type II fuzzy set modeling uncertainty; and the inverse steady-state prediction further enhances the steady-state adjustment capability of the system, reduces the control overshoot and steady-state error and ensures the stability and accuracy of temperature regulation [36,37]. Additionally, the interpretability of SHAP (SHapley Additive exPlanations) was used to analyze the degree of influence of the control parameters of the top temperature and top pressure of the atmospheric column of methanol distillation. By accurately adjusting the key parameters of the pressurized column’s top pressure, top temperature, feed flow rate and reflux ratio, it is possible to optimize the distillation process, ensure production efficiency and optimize the purity and quality of the final product.

The main contributions of this work are as follows:A novel methanol distillation control system is designed to effectively improve the stability and accuracy of the control by combining type II fuzzy neural networks, generative adversarial networks (GANs), and inverse steady-state prediction to address the complex production dynamics and long-term time-dependent problems in the methanol distillation industrial process.The model uncertainty of the second-class fuzzy set is introduced into the neural network structure to construct a fuzzy neural network. Combined with the general model control method, by analyzing the actual operating parameters of the pre-distillation column, pressurized column and atmospheric column, the key parameters, such as the temperature and pressure at the top of the atmospheric column, are effectively adjusted.The parameters of the methanol distillation model were analyzed in depth by using the SHAP method, and the influence degree of each parameter (such as the top pressure of the pressurized column, the feed parameters of each column and the reflux ratio) on the top temperature and top pressure of the atmospheric column was clarified. The internal working principle of the model is revealed from the mechanism level, which provides a theoretical basis for model optimization.The advantages of the combination of the GAN and T2FNN are mainly reflected in the data-driven control optimization, the improvement of control accuracy and generalization ability and the reduction in modeling errors. The GAN generates high-quality input data by learning the characteristics of historical data, which makes the T2FNN more efficient in learning the nonlinear relationship, and effectively avoids modeling errors due to low quality of data. At the same time, the augmented data generated by the GAN can supplement the shortcomings of the actual data, so that the T2FNN has higher prediction accuracy and stronger generalization ability under complex working conditions.

The rest of the paper is as follows: Section 2 introduces the process of methanol distillation in three towers. Section 3 describes the multi-sensor data fusion modeling for methanol distillation, including GAN data preprocessing, the steady-state inverse prediction strategy based on the type II fuzzy neural network, and the T2FNN model training method. Section 4 discusses the model comparison training results, methanol distillation comparison control experimental results and SHAP-based interpretability analysis. Finally, Section 5 provides the conclusions of the study, demonstrating the superiority of the GAN-T2FNN model in methanol distillation control, as well as the wide range of application prospects, and identifies future directions of work.

## 2. Background

Industrially, crude methanol is produced by reacting synthesis gases (CO and Hz) as reactants with CuO, ZnO, Al_2_O, etc., as catalysts under high-temperature catalysis. With the aid of a copper-based catalyst, high hydrogen from the decarbonization section reacts reversibly with carbon dioxide (CO_2_), carbon monoxide (CO) and hydrogen (H_2_) in the recycled gas to synthesize methanol (CH_3_OH) and water (H_2_O). The main reaction equations are as follows:(1)$$\mathrm{CO}+H_{2}\to {\mathrm{CH}}_{3}\mathrm{OH}$$(2)$${\mathrm{CO}}_{2}+H_{2}\to {\mathrm{CH}}_{3}\mathrm{OH}+H_{2}O$$

In the process of synthesizing methanol, the reaction between carbon dioxide (CO_2_) and hydrogen (H_2_) in coke oven gas produces a large amount of water (H_2_O), as well as many by-products, which directly affect the purity and quality of methanol, so separating and removing the water and by-products is the most important thing to do to obtain refined methanol. The by-products in methanol synthesis can be classified into light and heavy components according to their boiling points. Taking advantage of the difference in boiling points between these components and methanol, the three-tower distillation process partially gasifies and cools them several times, separating each component of the crude methanol, so as to obtain the refined methanol product that meets the industrial standard.

The three-column distillation process consists of a pre-distillation column, a pressurized column and an atmospheric column, as shown in Figure 2. Among them, the light component is completed in the distillation process of the pre-distillation tower, and the separation of the heavy component is completed by the distillation process of the pressurized tower and the atmospheric tower together [38].

The main function of the pre-distillation tower is to remove the higher content of dimethyl ether and some other impurity gases in the crude liquid. Crude methanol is first mixed with alkaline solution in the pre-distillation tower, where most of the light components are removed, and ether is recovered by using the condensing device at the top of the tower, while the non-condensable gases are discharged through the torch, and the condensate is pumped out from the bottom of the pre-distillation tower, and then pressurized by pumps to enter the pressurized tower to distill.

The pressurized tower adopts heat integration technology to exchange heat between the steam at the top of the pressurized tower, which has a temperature of about 122 °C, and the kettle liquid of the atmospheric tower. After heat exchange, the steam is condensed into liquid, part of which is refluxed back to the pressurized tower, and part of which is extracted as refined methanol products, and the kettle discharge liquid serves as the feed material for the subsequent atmospheric tower.

In the atmospheric pressure tower, methanol, light and heavy components and water can be completely separated. The gas phase at the top of the tower is the methanol vapor containing traces of non-condensable gas, and the refined methanol product is obtained after condensation. The side line of the lower part of the tower extracts heterohydric alcohol oil as the feed material of the recovery tower, and the discharge liquid from the tower kettle is water containing trace methanol, which can be sent to the sewage treatment plant for treatment [39].

## 3. Multi-Sensor Data Fusion Modeling Based on GAN-T2FNN

The methanol distillation production process is a complex control process that utilizes a large number of different kinds of sensors for signal acquisition. The main characteristics of these extracted signals are complex and nonlinear and have high dimensionality. The use of deep learning modeling can provide excellent data fitting, noise filtering, multi-sensor fusion, analysis and prediction. The GAN is utilized to preprocess the data collected from each sensor to solve the problem of low data volume and uneven data quality. The T2FNN model established on the basis of the inverse steady-state fuzzy neural network is trained according to the preprocessed data to realize the multi-sensor data fusion and achieve the purpose of accurate control and prediction of key parameters.

### 3.1. The GAN Performs Numerical Preprocessing

In the process of methanol purification, data prediction for pressurized and atmospheric towers is important in controlling the feedstock percentage and correcting the facility parameters. A large amount of labeled training data is required to apply complex deep learning models well to industrial data prediction problems. As in real production, the main problems faced in data collection are poor receiver sensitivity, a high failure rate and improper operation by workers. Therefore, the amount of relevant data in real industry is generally low and the data are of low quality. The lack of training data leads to a decrease in the effectiveness of training the network. To reduce the dependence of the model on the accuracy and quantity of data, a GAN model is used to learn the characteristics of the data of the existing industrial demand so that training samples can be generated to match its distribution. The GAN is a model based entirely on adversarial training and was proposed by Ian Goodfellow et al. in 2014. The core idea of the GAN is the Nash equilibrium from game theory, and the method is mainly composed of two parts: one is the generator and the other is the discriminator. The overall process is shown in Figure 3.

The generator receives random noise data $p_{z}(z)$ that follows a probability distribution z as input and produces synthetic data samples G(z) that also adhere to the same probability distribution $p_{G}(z)$. The input for the discriminator comes from both the generated samples G(z) and the real samples x (where x is derived from the probability distribution $p_{data}(x)$). The output of the discriminator D is a scalar value D(G(z)) that indicates the probability that the generated sample G(z) conforms to the real distribution $p_{data}(x)$. When the generator G is fixed, the optimization of the discriminator is similar to the training of a conventional binary classifier, and its objective function can be represented using cross-entropy:(3)$$J\left(D\right)=-\frac{1}{2}E_{x∼p_{\mathrm{data}}}\left[\log D\left(x\right)\right]-\frac{1}{2}E_{x}\log\left[1-D\left(G\left(Z\right)\right)\right]$$

In the equation, D and G represent the differentiable functions of the generator and discriminator, respectively. The term G denotes the expected distribution, while x refers to the real data samples, and z is the random noise vector. The term G(z) signifies the data generated by the discriminator. The first term of Equation (1) indicates that D labels the real data x as 1, while the second term shows that D labels the generated data G(z), which is produced by mapping the noise vector z through the generator G, as 0. This leads to the derivation in Equation (3).(4)$$D_{G}^{*}\left(x\right)=\frac{p_{data}\left(x\right)}{p_{data}\left(x\right)+p_{g}\left(x\right)}$$

The training objective of the generator is to make the probability distribution $p_{G}(z)$ (Gaussian distribution) of the generated samples G(z) as close as possible to the probability distribution $p_{data}(x)$ of the real samples x.

The objective of the discriminator is to accurately classify the input and optimize the neural network through gradient feedback. Specifically, when the input is real data x, the goal is to make the output probability value D(x) as close to 1 as possible; conversely, when the input is G(z), the aim is to make D(G(z)) approach 0. Therefore, we can formalize the expression of the discriminator’s objective function as follows:(5)$$J\left(D\right)=\max\left\{E_{x∼p_{\mathrm{data}}}\left[\log D\left(x\right)\right]+E_{z}\log\left[1-D\left(G\left(z\right)\right)\right]\right\}$$

During the adversarial training process, the generator and discriminator achieve mutual optimization, reaching a Nash equilibrium. When the discriminator is unable to effectively distinguish the inputs, it indicates that the generator has learned a probability distribution similar to that of the real samples.

According to the training and optimization objectives of the generator G and discriminator D, the loss functions for the generator and discriminator are denoted as $L_{G}$ and $L_{D}$, respectively:(6)$$L_{G}=-E_{z∼p_{z}\left(z\right)}\left\{\log\left[1-D\left(G\left(z\right)\right)\right]\right\}$$(7)$$L_{D}=-E_{z∼pdata\left(x\right)}\left\{\log\left[D\left(x\right)\right]\right\}+E_{z∼p_{z}\left(z\right)}\left\{\log\left[1-D\left(G\left(z\right)\right)\right]\right\}$$

In this equation, E represents the expected value of the distribution. Based on Equations (4) and (5), we can formulate the objective function for the GAN:(8)$$\underset{G}{min}\underset{D}{max}V\left(D,G\right)=E_{x∼pdata\left(x\right)}\left[\log D\left(x\right)\right]+E_{z∼p_{z}\left(x\right)}\left\{\log\left[1-D\left(G\left(z\right)\right)\right]\right\}$$

In this objective function, V(D,G) represents a binary cross-entropy function. The ultimate goal of this function is to align the generated sample probability distribution $p_{G}(z)$ with the true sample probabilities. This approach facilitates the generation of data that conform to the characteristics of the original dataset, thereby aiding in the training of neural networks for steady-state inverse prediction.

### 3.2. Steady-State Inverse Prediction Based on Type II Fuzzy Neural Networks

The methanol distillation process is a complex industrial procedure characterized by intricate interactions among various columns. This paper focuses on the control of the atmospheric pressure column, aiming to achieve effective regulation of key parameters, specifically the top temperature and top pressure of the column, by analyzing the actual operational parameters of the pre-distillation column, pressurized column and atmospheric pressure column. In this process, the steady-state inverse modeling based on a neural network is combined with the general model control method to give the control strategy of the multi-input multi-output system, as shown in Figure 4.

The GMC (Generalized Minimum Variance Control) method assumes that the process exhibits first-order dynamic characteristics and calculates the steady-state target values of the controlled outputs (*XD*; *ss* and XB; *ss*) as follows:(9)$$Y_{x}=Y+K_{1}^{p}\left(Y_{p}-Y\right)+K_{2}^{p}\int_{0}^{t}\left(Y_{p}-Y\right)dt+b_{y}\;X_{ss}=X+K_{1}^{t}\left(X_{sp}-X\right)+K_{2}^{t}\int_{0}^{t}\left(X_{sp}-X\right)dt+b_{x}$$
where $Y_{sp}$ and $X_{sp}$ are the desired set values of Y and X, $K_{1}^{p}$; $K_{2}^{p}$; $K_{1}^{t}$ and $K_{2}^{t}$ and the control law tuning constant. Deviations $b_{y}$ and $b_{x}$ represent steady-state mismatches between the process and the neural network model and are calculated only once when the controller is switched from “manual” (open-loop) mode to “automatic” (closed-loop) mode.(10)$$b_{y}=Y_{ss}-Y\;b_{x}=X_{ss}-X$$

In addition, when switching to automatic mode, the calculated setpoint is consistent with the last measured value of the process variable. When the controller is in manual mode, its integral is reset to zero. The nonlinear controller reads the process variables at each control interval and calculates the target values $Y_{ss}$ and $X_{ss}$ according to the corresponding equations. Then, the steady-state target value and the actual measured values of the pre-rectification tower, pressurized tower and atmospheric tower are taken as the input of the neural network model of the rectification tower, and then, the top temperature and top pressure are calculated by the network, so as to promote the whole process to reach the temporary steady-state target $Y_{ss}$ and $X_{ss}$. When the perturbation change can be measured, the perturbation change can be transmitted directly to the model, so that the neural network controller can provide a nonlinear feedforward response. On the contrary, if the disturbance change cannot be measured, the controller will use feedback to eliminate the unmeasured interference, and then complete the whole process of the steady-state inverse prediction of methanol rectification.

### 3.3. Model Training for T2FNN

After preprocessing the data through the GAN, the methanol distillation control system is trained using past data. In this paper, we make use of the fact that type II fuzzy sets can model both intra-individual uncertainty and inter-individual uncertainty, i.e., additional uncertainty is introduced on top of type I fuzzy sets, and its affiliation function $\mu_{A}(x)$ is no longer a single value but a fuzzy set, i.e., the affiliation corresponding to each input value x is itself a fuzzy affiliation function, denoted as $\mu_{A}\left(x,u\right)$, where u is the type II affiliation degree defined on [0, 1] [40,41]. It can be expressed as Equation (11):(11)$$A=\left\{\left(x,u,\mu_{A}\left(x,u\right)\right)|x\in X,u\in J_{x}\subseteq \left[0,\;1\right]\right\}$$
where x∈X is the elements of the fuzzy set; $u\in J_{x}\subseteq [0,1]$ is the range of values of the degree of affiliation; $\mu_{A}(x,u)\in [0,1]$ is the type II degree of affiliation.

However, due to the high complexity of the general type II fuzzy set computation, a simplified interval type II fuzzy set is usually used, where the degree of affiliation of the interval type II fuzzy set is no longer a fuzzy set but an interval $\left[\underline{\mu }(x),\bar{\mu }(x)\right]$, $\underline{\mu }(x)$, $\bar{\mu }(x)$ denoting the upper and lower bounds of the degree of affiliation of the input value x, respectively [42]. The interval type II fuzzy set can be expressed by the following formula:(12)$$A=\left\{\left(x,\left[\mu \left(x\right),\bar{\mu }\left(x\right)\right]\right)|x\in X\right\}$$

The core idea of the fuzzy neural network is to introduce the inference mechanism of fuzzy logic into the architecture of the neural network, and type II fuzzy sets are utilized to make it more robust and adaptive. The network structure is shown in Figure 5.

At the input layer, the fuzzy neural network first fuzzifies the input. Assuming that the input is x, the fuzzification converts the input to a fuzzy affiliation value through an affiliation function. The affiliation function used in this paper is a Gaussian function and is formulated for the lower and upper affiliation functions for a type II fuzzy layer:(13)$$\begin{array}{c} \mu_{lower}\left(x\right)=\exp\left(-{\left(\frac{x-\mu_{lower}}{\sigma_{lower}}\right)}^{2}\right)\\ \mu_{upper}\left(x\right)=\exp(-\left(\frac{x-\mu_{upper}}{\sigma_{upper}}{)}^{2}\right) \end{array}$$
where μ is the mean of the Gaussian function and σ is the standard deviation, which controls the width of the affiliation function. The fuzzy output values are expressed by averaging the combined lower and upper affiliation limits to capture the uncertainty within the fuzzy set boundaries.(14)$$\mu \left(x\right)=\frac{\mu_{lower}\left(x\right)+\mu_{upper}\left(x\right)}{2}$$

Each fuzzy rule calculates the value of the input’s affiliation to different rules by means of a fuzzy affiliation function. In the middle layer of the model, the fuzzy rule base contains multiple fuzzy rules, and each rule is reasoned based on the fuzzy affiliation of the inputs and by adjusting the value of the fuzzy output. The formula is given below:(15)$$\omega_{j}=\overset{n}{\underset{i=1}{\prod }}\mu_{A}\left(x_{i}\right)$$
where $\omega_{j}$ denotes the weight of each fuzzy rule and $\mu_{A_{i}}(x_{i})$ denotes the affiliation of the input $x_{i}$ to the fuzzy set $A_{i}$. This weight determines the activation level of the rule.

The last layer of the fuzzy neural network is the defuzzification layer, whose task is to convert the fuzzy inference results into specific output values. The commonly used defuzzification method is the weighted average method with the following formula:(16)$$y=\frac{\sum_{j=1}^{m}w_{j}\cdot y_{j}}{\sum_{j=1}^{m}w_{j}}$$

Throughout the training process of the model, the parameters of the fuzzy affiliation function (μ and σ) and the network weights are updated by the back-propagation algorithm, which ultimately realizes the efficient processing and modeling of complex data.

## 4. Results

The experimental data in this study were obtained from the actual production data of the Hebei Methanol Plant in China. The data include the extracted data from various types of sensors in the pre-distillation tower, pressurized tower and atmospheric tower, such as the top temperature, top pressure and extracted moisture. The production plant uses the traditional PID control method with a total of 2000 pieces of data and data preprocessing for the removal of outliers in the collection of sensor data. At the same time, the control of the entire control system was also recorded, so as to facilitate the subsequent comparative experiments to analyze the results. The hardware configuration used for the experiment includes an AMD Ryzen 7 5800H processor (AMD, Santa Clara, CA, USA), an NVIDIA GeForce RTX 3060 GPU (NVIDIA, Santa Clara, CA, USA), and 32GB DDR4 3200MHz memory to provide sufficient computing power and data storage space. In terms of the software environment, the experiments were run based on the Windows 11 operating system and used the PyTorch 2.0.1 deep learning framework, combined with CUDA 11.8 for GPU acceleration, and Python 3.8.10 as the programming language. Table 1 shows some of the training parameters of the model.

In this study, real data in the industrial production process were collected, and the experimental data were divided into a training set, testing set and validation set with the ratio of 8:1:1. MAE, MSE, RMSE and R^2^ were used to judge the hybrid model, and the formula for each evaluation criterion is as follows: (17)$$\mathrm{MAE}=\frac{1}{n}\overset{n}{\underset{i=1}{\sum }}\left|y_{i}-{\hat{y}}_{i}\right|\;\mathrm{MSE}=\frac{1}{n}\overset{n}{\underset{i=1}{\sum }}\left|y_{i}-{\hat{y}}_{i}\right|\;\mathrm{RMSE}=\sqrt{\frac{1}{n}\overset{n}{\underset{i=1}{\sum }}{\left(x_{i}-{\hat{x}}_{i}\right)}^{2}}\;\;R^{2}=1-\frac{\sum_{i=1}^{n}{\left(y_{i}-{\hat{y}}_{i}\right)}^{2}}{\sum_{i=1}^{n}{\left(y_{i}-{\bar{y}}_{i}\right)}^{2}}$$
where n is the number of samples, $x_{i}$ is the measured value of the first i measured value of the first sample, ${\hat{x}}_{i}$ is the predicted value, i is the predicted value of the first sample and x is the average of the samples.

To enhance the persuasiveness of the experimental conclusions, we conducted two separate experiments using the training set. One type was a comparison experiment carried out for different neural network models. The other type was an ablation experiment conducted on the GAN-T2FNN model itself, which verifies the superiority shown by the combination of the GAN and T2FNN in terms of control.

### 4.1. Comparative Experimental Results of Different Neural Network Models Training

We utilized commonly used regression model evaluation metrics to measure the performance of different neural network models in control system experiments. The evaluation indexes of each model obtained through calculation are shown in Table 2 and Table 3.

The predictions of the different models can be visualized very well in Figure 6. Overall, the GAN-T2FNN network outperforms the LSTM and BP neural network models in all categories of metrics. Analyzed in terms of MAE, MSE and RMSE, the smaller metrics of the GAN-T2FNN network indicate that the prediction is more accurate and more sensitive to abnormal data. The value of the R^2^ Score, on the other hand, reflects the greater interpretability of the GAN-T2FNN model.

By comparing the performance of different models in the prediction of temperature and pressure at the top of an atmospheric tower, it is easy to find that the GAN-T2FNN model is superior in performance, both in the modeling of complex distributions and in accurate prediction. Since the GAN is capable of augmented learning of data features, the T2FNN combined with it performs better in handling complex feature data and multi-sensor data fusion, mainly in terms of improved prediction accuracy and data fitting performance. This performance improvement is significant for the control of all aspects of this process.

### 4.2. Results of Ablation Experiments for GAN-T2FNN Model Training

The evaluation indexes of each model obtained through calculation are shown in Table 4 and Table 5.

As shown in Figure 7, in terms of MAE, MSE and RMSE, the GAN-FNN and GAN-T2FNN prediction models have higher prediction accuracy than the FNN and T2FNN models. In terms of square coefficient R, the prediction results of the GAN-FNN and GAN-T2FNN models fit the actual values well. The prediction error and anomaly values are small, and the models can better reflect the trend of data change.

The combination of the GAN algorithm improved the model’s performance in complex distribution modeling, feature learning and prediction accuracy. Compared with the traditional type I FNN model, the type II FNN model is superior in terms of nonlinear mapping ability, convergence speed, overfitting reduction, etc. The model’s influence on MAE, MSE, RMSE and R^2^ is also reflected. In addition, the GAN-T2FNN model adds a generative adversarial network on the basis of the T2FNN. GAN-T2FNN is especially good at processing data with noise interference or complex features. With the enhanced feature learning ability, the model can better capture the intrinsic pattern of the data, so as to optimize the prediction effect. Therefore, it can be concluded that the participation of the GAN algorithm can improve the overall effect of the type 2 FNN model more, and significantly improve the performance of data prediction and fitting. The GAN-T2FNN model obviously improves data prediction and fitting performance, which can better improve the prediction accuracy and provide more accurate and scientific predictions for methanol rectification production, which plays an important role in the subsequent improvement of the methanol purification accuracy and material utilization rate control process.

### 4.3. Methanol Distillation Control Experiment

The above-designed methanol distillation control system was applied to the actual production of the plant to verify its actual performance. In the experimental process, we trained the model using the corresponding data from the pre-distillation tower, pre-pressure tower and atmospheric pressure tower collected by a series of sensors in the methanol distillation plant. Then, the core indicators of methanol distillation, the maximum temperature and pressure of methanol distillation and the purity of the final product were recorded in detail, and the fluctuation of the whole distillation process was closely tracked. Through visual analysis of the data, it can be clearly observed that the system performance after optimization was significantly improved. By visually analyzing the data in Table 6, it can be clearly observed that the performance of the optimized system has been significantly improved.

From the Figure 8 curve, it can be seen that for the top temperature of the atmospheric tower, the fluctuation of the PID control was the most drastic, showing a high sensitivity to external perturbations, and the ability to adapt to complex working conditions was poor. As is also shown in the table, its variance was the highest (6.94 × 10^−6^), which indicates that there is a large instability of the system in the operation process, which leads to difficulty in attaining a consistent product quality of different batches of the product. The volatility of the BP neural network control improved, but there were still large fluctuations; its variance was 4.25 × 10^−6^. The stability of LSTM control was relatively good; the variance was 3.59 × 10^−6^, but there were still some fluctuations, they were smaller due to the stable performance of GAN-T2FNN. The control curve of GAN-T2FNN was the smoothest, with small fluctuations, with the lowest variance (2.37 × 10^−6^) and the smallest fluctuation range (0.006), showing very high anti-interference ability, corresponding to the low variance and small fluctuation range shown in the table, which further indicates that the model optimizes the shortcomings of the traditional control methods, and greatly improves the control accuracy and system robustness.

From the Table 7, it can be seen that GAN-T2FNN outperforms the other models in all key indicators. In terms of temperature control at the top of the tower, the mean value of GAN-T2FNN was 51.146 °C, which is within a more desirable range. In terms of variance, GAN-T2FNN had the lowest variance, of 0.348249, which indicates that its volatility is extremely small and that it can better adapt to changes in the external environment, while the variance of PID control was as high as 1.21114, indicating that its system is more volatile and not stable enough. Regarding the fluctuation range of the tower top temperature, the fluctuation range of GAN-T2FNN was only 2 °C, while the fluctuation ranges of LSTM, the BP neural network and PID control were 3.44 °C, 3.79 °C and 4.1 °C, respectively, which indicates that the fluctuation of the traditional control methods is larger and that there are more unstable factors.

From Figure 9, it can be seen that compared with the GAN-T2FNN model, the shortcomings of the traditional PID control and other common neural network models are relatively obvious. The PID control had the largest fluctuation in the tower top temperature, between 48.1 °C and 52.15 °C. The LSTM and BP neural networks had about the same range of fluctuations. The LSTM and BP neural networks, on the other hand, had essentially the same range of fluctuations, remaining between 48 °C and 52 °C. The other three control models had the most extreme fluctuations. Obviously, the other three control models had larger extremes, i.e., more anomalous data, indicating that the robustness of the system is relatively poor, while the GAN-T2FNN optimization model is more stable and more resistant to interference.

The GAN-T2FNN model significantly reduced the fluctuation in the moisture content of the methanol products and improved the stability of production and product quality by optimizing the top temperature and pressure control of an atmospheric tower. Based on Table 8, the traditional PID control and LSTM and BP neural networks had large fluctuations when controlling the moisture content of methanol products, especially the PID control, which had a mean value of 11.42, a variance of 4.50, and a fluctuation range of 8.11, indicating that its control accuracy is low, and the quality difference between batches was large, which makes it difficult to ensure the stable moisture content of the products. The BP neural network had a control effect that was better than PID, but still had large fluctuations, and its quality varied greatly between batches, which makes it difficult to ensure a stable moisture content of the products. Although the control effect of the BP neural network was better than that of PID, there was still a large fluctuation, with a mean value of 12.03, a variance of 2.16 and a fluctuation range of 5.10, which indicates that the system is still not stable enough in the face of disturbances. Compared with the BP neural network and PID control, LSTM could better maintain the product quality, but there were still some problems with fluctuation. The GAN-T2FNN model, on the other hand, showed the best control effect, with a mean value of only 10.79, a minimum variance of 1.11 and a minimum fluctuation range of only 3.51, which indicates that the model has a stronger stability and anti-disturbance ability when dealing with complex working conditions.

By improving the control of the top temperature and pressure of the atmospheric pressure tower, the moisture content of the extracted methanol was significantly reduced, as shown in Figure 10. In order to ensure the quality of the products, the water content of the methanol extracted (moisture content of finished products) should be maintained at about 10%. The moisture content of the finished products fluctuated from 10% to 15% under the LSTM and BP neural network control models, while the moisture content of the finished products fluctuated the most under the traditional PID control, which means that the quality of the products from different batches of the products varied more seriously. After the optimization of the GAN-T2FNN model, the moisture content of most of the products in different batches was 10%, and it basically maintained a slight fluctuation between 10% and 12%, which indicates that the product quality met the standard and the production quality was stable under the control of the GAN-T2FNN model.

Analyzing and comparing the experimental results from a global perspective, the instability of the external environment (ambient temperature and humidity), the lack of accuracy of the sensors and the temperature of the cooling water all affected the experimental results. The optimized control model still had slight fluctuations in response to force major factors, but the fluctuation amplitude was significantly reduced compared to other control models, showing stronger stability and robustness. From the above three core indexes of tower top temperature, pressure and methanol water content, the sharp contrast and significant difference between before and after optimization strongly confirm the outstanding advantages of the methanol control system constructed in this paper, which performed well in ensuring the stability of product quality and improving the distillation efficiency.

### 4.4. Interpretability Analysis Based on SHAP

In order to clearly express the prediction of the GAN-T2FNN model for important indicators and parameters, SHAP (SHapley Additive exPlanations) was used to analyze its interpretability. We characterize the extent of this effect by the SHAP value of each parameter, where a positive value (on the right side of the plot) implies that the feature contributes positively to the model output, i.e., an increase in the value of the feature leads to an increase in the value of the model output, and a negative value (on the left side of the plot) implies that the feature contributes negatively to the model output, i.e., an increase in the value of the feature leads to a decrease in the value of the model output. Figure 11 and Figure 12 both show swarm diagrams, with the vertical axis representing the main production parameters and the horizontal axis representing the SHAP eigenvalues of these parameters, changes in which directly affect the output.

Figure 11 summarizes the predicted SHAP values for each parameter of the methanol distillation model for the top pressure of the atmospheric tower. Macroscopically, it shows that pressurized tower top pressure is the most important influencing factor. This influence is multifaceted and has complex causes. The most important reason for this is that since the pressurized tower is connected to the atmospheric tower through the gas phase pipeline, once the pressurized tower top pressure changes, it will affect the flow rate of the gas phase in the pipeline, which in turn will change the amount of gas transferred to the atmospheric tower, resulting in a change in the atmospheric tower top pressure [43]. At the same time, the pressurized tower top pressure change will also affect the heat load and energy transfer of the system, as well as the material balance, which will lead to changes in the operating conditions of the atmospheric tower, and thus indirectly affect the atmospheric tower top pressure [44].

The two characteristics of atmospheric tower feed and pressurized tower feed are also important factors affecting the pressure at the top of the atmospheric tower. The causes of this effect are mainly composed of three aspects: feed volume, feed temperature and feed composition [45]. The amount of feed directly affects the level of liquid in the tower, and the feed temperature affects the temperature in the tower, which in turn affects the pressure at the top of the tower. An increase in the recombinant content of the feed will lead to an increase in the pressure at the top of the tower, and the reverse will cause a decrease in the pressure. At the same time, these two characteristics also indirectly affect the parameters, such as reflux ratio and extraction volume, which will lead to changes in the top pressure of the atmospheric tower.

As for the feature of extracting ethanol content from an atmospheric tower, its predicted impact on the output is mainly negative. This is because the evaporation of ethanol increases the gas phase load in the tower, which affects the top pressure. When the ethanol content extracted from an atmospheric tower increases, the top pressure of the tower may rise if other conditions remain unchanged [46].

Among the features related to pre-distillation columns, the fluctuation in pressure at the top of the pre-distillation column can disrupt the material balance and gas–liquid equilibrium, which affects the pressure stability of the subsequent atmospheric column. And, the influence of the feed side of the pre-distillation tower is mainly due to the change in the feed flow rate affecting its heat load, which can cause pressure fluctuations in the subsequent tower [47].

Other characteristic parameters of the pressurized tower, such as the top temperature and reflux ratio and the influence on the top temperature of the atmospheric pressure tower, are manifested in the following: an increase in the top temperature of the pressurized tower may increase the gas phase load in the tower, which will lead to a rise in the pressure at the top of the atmospheric pressure tower if the subsequent treatment is not timely. Increasing the reflux ratio can reduce the top temperature of the tower, reduce the gas phase load and help reduce the top pressure of the atmospheric pressure tower. However, a reflux ratio that is too large may also lead to excessive loading in the tower, instead of causing pressure rise [48]. Of course, fluctuations in other parameters will also have a certain effect on the top pressure of the atmospheric pressure tower, but the extent of these effects is small.

Figure 12 summarizes the influences of each parameter on the top temperature of the autoclave tower. In this part of the SHAP value analysis, the SHAP value of the pressurized tower top pressure remains the most critical influencing feature. The change in pressurized tower top pressure will directly affect the gas phase temperature, which in turn will have an impact on the temperature of the atmospheric tower. When the pressurized tower top pressure is insufficient, the tower top temperature will be reduced, resulting in an insufficient heat load of the atmospheric pressure tower, which, in turn, reduces the atmospheric pressure tower top temperature. At the same time, the change in pressurized tower top pressure also affects the gas phase load in the tower and the fraction of distillate at the top of the tower, which also indirectly affects the temperature of the autoclave [49].

The influence of atmospheric tower feed and pressurized tower feed on the top temperature of the atmospheric tower is also very significant. The feeding position and feeding temperature of the atmospheric pressure tower are important factors affecting the temperature at the top of the tower. An appropriate feed location and feed temperature can maintain the thermal balance in the tower and ensure that the temperature at the top of the tower is stable [50]. The feed to a pressurized tower usually contains higher concentrations of volatile components, such as ethanol or other organic liquids. With the entry of these components, a change occurs in the composition of the vapor phase in the atmospheric tower, which affects the temperature of the gas phase. Moreover, the increase in volatile components in the feed causes the vapor pressure in the tower to be enhanced, changing the thermal balance at the top of the tower and leading to an increase in the temperature at the top of the tower [51].

The SHAP values of the two parameters, pre-distillation tower feed and pre-distillation tower top pressure, are relatively close to each other, and they have a significant effect on the top temperature of the atmospheric tower. A higher feed flow rate may lead to a rise in the liquid level in the tower and a decrease in the gas–liquid contact time, which affects the evaporation and condensation process of the gas, thus leading to the instability of the temperature at the top of the tower in the subsequent process, which may result in an increase or decrease in the temperature. And, the increase in pressure of the pre-distillation column may bring light components into the atmospheric column and affect the reboiler heat supply of the atmospheric column, thus affecting the top temperature of the atmospheric column and the quality of methanol [52,53].

Through the comparison of SHAP values, the influence of the two sets of characteristic values, atmospheric tower reflux ratio and pressurized tower reflux ratio, on the top temperature of the atmospheric tower cannot be ignored. An increase in the reflux ratio of the atmospheric tower will lead to a rise in the liquid level in the tower, which may block the gas phase flow, causing some of the high-boiling components to fail to be effectively separated, leading to a decrease in the temperature at the top of the tower. The pressurized tower reflux ratio changes directly affect the inlet temperature of the atmospheric tower, which in turn causes a relevant change in the temperature at the top of the tower, and this fluctuation not only affects the material balance of the atmospheric tower, but also leads to changes in the pressure at the top of the tower [54].

Other features shown in Figure 12 have SHAP values that are often close to zero, indicating that their impact on the pressure vessel top temperature is very small, and they will not be further analyzed.

## 5. Conclusions

In this study, an innovative control strategy that deeply integrates a T2FNN, a GAN and an inverse steady-state prediction technique is proposed. The performance of the inverse steady-state prediction model based on the type II fuzzy neural network was greatly improved after the introduction of the GAN for data preprocessing. Compared using different model evaluation metrics, the GAN-T2FNN model was shown to have better performance in terms of prediction accuracy and fitting effect, with an MAE value of 0.1828 for better robustness, MSE and RMSE values of 0.0732 and 0.2706 for higher prediction accuracy, and an R^2^ Score of 0.9854, which is closer to 1, and it had the best overall performance among all models. By applying the optimized methanol distillation control system to the actual plant production, the overall performance of the system was significantly improved. Through the comparison test, the monitoring and analysis of key indexes showed that the fluctuation ranges of the temperature and pressure at the top of the atmospheric tower were greatly reduced, the stability was obviously enhanced, and the sudden change phenomenon was effectively suppressed. The moisture content of the methanol product was obviously reduced, and the purity and quality of the product were obviously improved. And, with the help of SHAP analysis, the influence mechanism of each parameter on the top temperature and pressure of the tower was revealed in depth. The GAN-T2FNN model can also be adapted to more complex or larger-scale industrial environments. The combination of a GAN and a T2FNN can effectively deal with large-scale datasets and mitigate the challenges of high-dimensional, sparse data and uncertainty through data generation and model optimization techniques. By integrating deep learning and a GAN, the control strategy can be adjusted in real time during the production process, reflecting the adaptability and responsiveness of the system. In addition to the methanol production process, the method can be applied to a number of other industrial processes with similar complexity and dynamics, such as the petroleum refining process, which usually contains multiple reactors, fractionating columns and other equipment, and whose reaction process is usually characterized by nonlinear, multiphase flow and time-varying characteristics. The GAN-T2FNN model can effectively control the temperature and pressure in the petroleum refining process to improve refining efficiency and product quality. Although the GAN-T2FNN model has a good optimization effect in methanol distillation, it may not be able to fully meet the requirements of millisecond response times due to the complex structure of the model. In order to cope with these challenges, future research can further explore the adaptive performance of the model under multiple, complex working conditions and try to incorporate a lightweight neural network structure or further increase the data enhancement techniques to improve the model’s adaptive capability. Its feasibility and effectiveness in large-scale chemical production can be further improved through a deep integration with IoT technology, edge computing and other cutting-edge technologies.

## Figures and Tables

**Figure 1 sensors-25-01308-f001:**
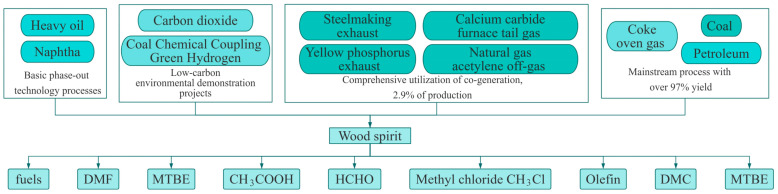
The production process of chemical raw materials and their downstream products.

**Figure 2 sensors-25-01308-f002:**
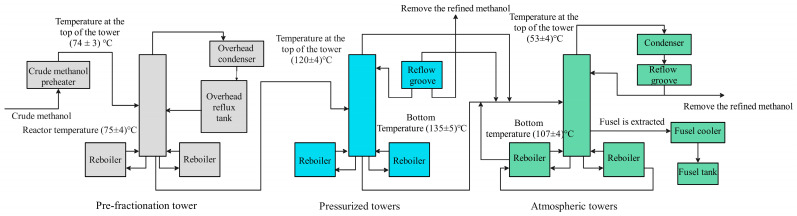
Methanol distillation process diagram.

**Figure 3 sensors-25-01308-f003:**
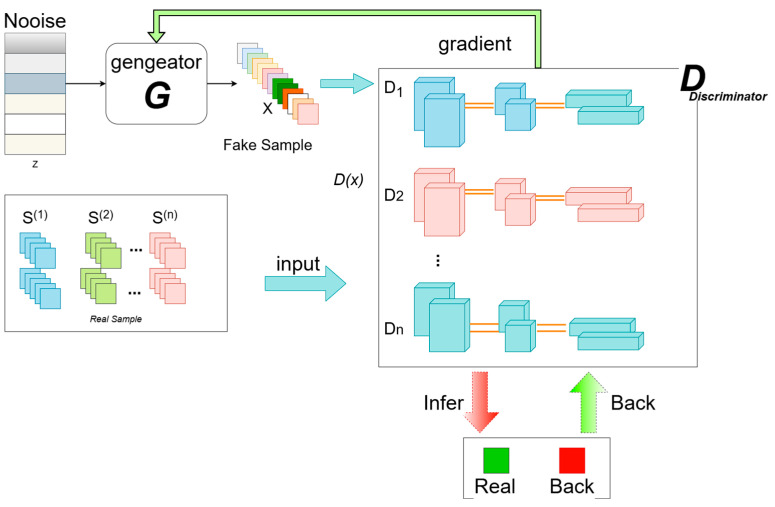
GAN architecture diagram.

**Figure 4 sensors-25-01308-f004:**
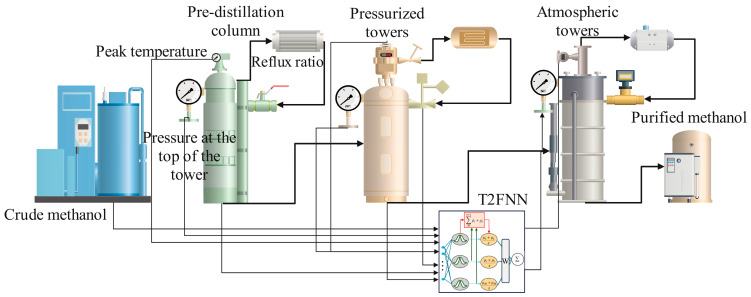
Inverse steady-state predictive control strategy based on type II fuzzy neural network models.

**Figure 5 sensors-25-01308-f005:**
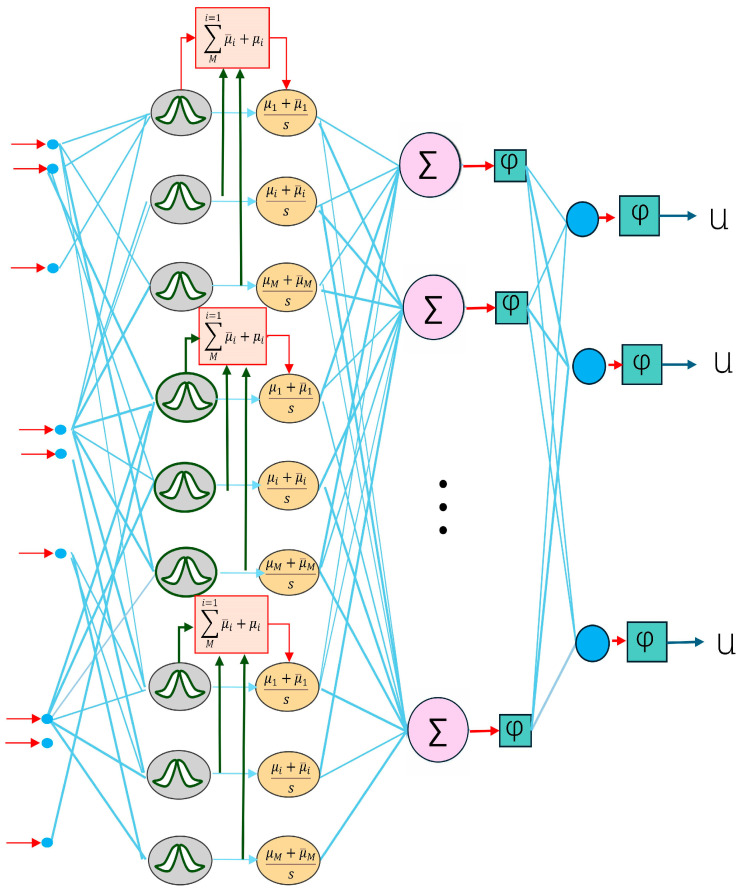
Schematic diagram of type II fuzzy neural network.

**Figure 6 sensors-25-01308-f006:**
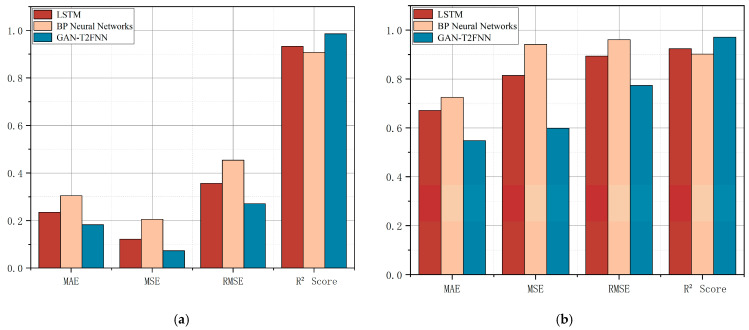
(**a**) Performance of different models for tower top temperature prediction. (**b**) Performance of different models for tower top pressure prediction.

**Figure 7 sensors-25-01308-f007:**
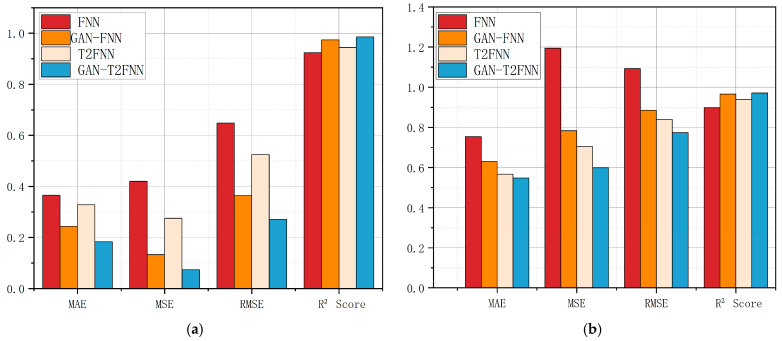
(**a**) Performance evaluation of the tower top temperature model; (**b**) performance evaluation of the tower top pressure model.

**Figure 8 sensors-25-01308-f008:**
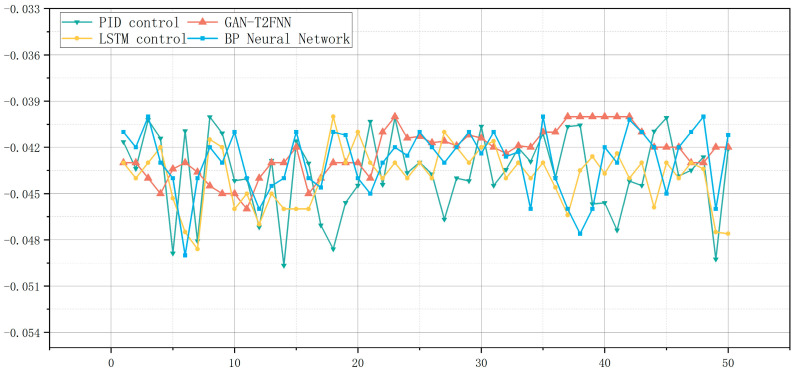
Top pressure of an atmospheric tower.

**Figure 9 sensors-25-01308-f009:**
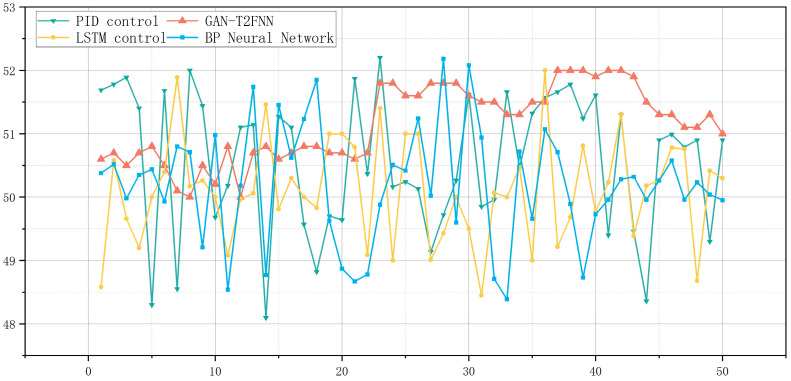
Top temperature of atmospheric tower.

**Figure 10 sensors-25-01308-f010:**
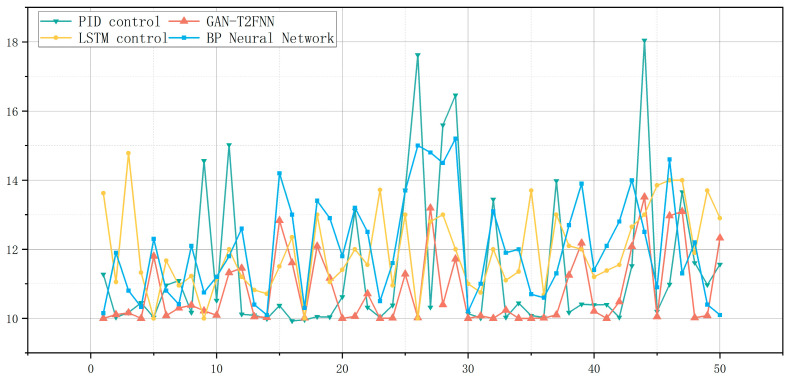
Thousandths ratio of water precipitated from an atmospheric tower.

**Figure 11 sensors-25-01308-f011:**
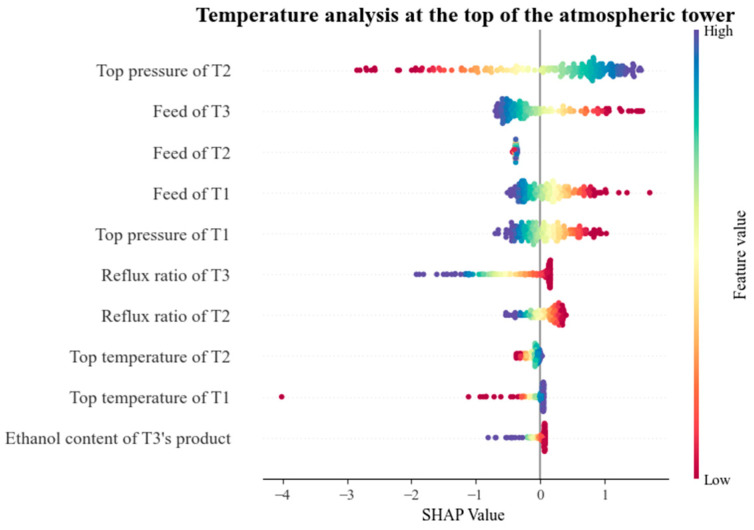
SHAP values of methanol overhead temperature in atmospheric columns (T1, T2 and T3 represent pre-distillation column, pressurized column and atmospheric column, respectively).

**Figure 12 sensors-25-01308-f012:**
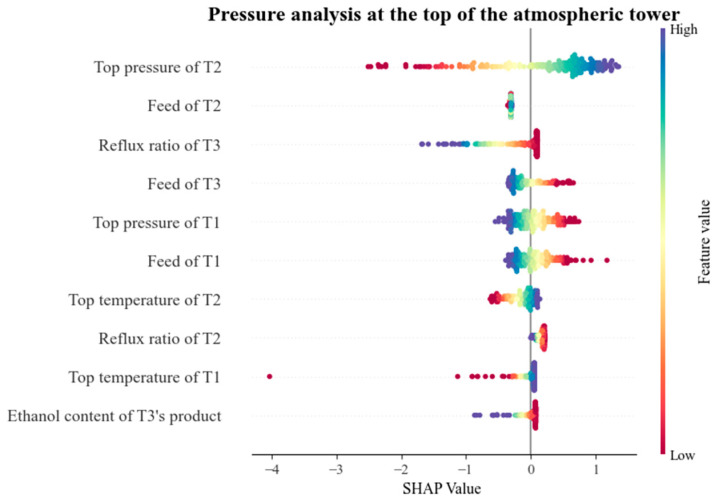
SHAP values of methanol overhead pressure in atmospheric columns (T1, T2 and T3 represent pre-distillation column, pressurized column and atmospheric column, respectively).

**Table 1 sensors-25-01308-t001:** Model parameters.

Layer Name	Input Dimension	Output Dimension
Full connectivity layer	output_dim	128
Type2FuzzyLayer	128	5 × 128
AuxiliaryLayer	5 × 128	5 × 128

**Table 2 sensors-25-01308-t002:** Comparison of the prediction results of different control models for the temperature at the top of an atmospheric pressure tower.

	LSTM	BP Neural Network	GAN-T2FNN Network
MAE	0.235	0.305	0.1828
MSE	0.122	0.205	0.0732
RMSE	0.3568	0.4543	0.2706
R^2^ Score	0.933	0.908	0.9854

**Table 3 sensors-25-01308-t003:** Comparison of the prediction results of different control models for the top pressure of an atmospheric tower.

	LSTM	BP Neural Network	GAN-T2FNN Network
MAE	0.672	0.725	0.1828
MSE	0.815	0.942	0.0732
RMSE	0.8938	0.961	0.2706
R^2^ Score	0.924	0.902	0.9854

**Table 4 sensors-25-01308-t004:** Comparative presentation of experimental results for prediction of tower top temperature in an atmospheric tower.

	FNN	GAN-FNN	T2FNN	GAN-T2FNN
MAE	0.3653	0.243	0.3275	0.1828
MSE	0.4201	0.1322	0.2748	0.0732
RMSE	0.6482	0.3636	0.5243	0.2706
R^2^ Score	0.9238	0.974	0.9438	0.9854

**Table 5 sensors-25-01308-t005:** Comparative presentation of experimental results for prediction of tower top pressure in an atmospheric tower.

	FNN	GAN-FNN	T2FNN	GAN-T2FNN
MAE	0.7536	0.6304	0.5665	0.5481
MSE	1.1941	0.7831	0.7046	0.5985
RMSE	1.0928	0.8849	0.8394	0.7736
R^2^ Score	0.8981	0.9657	0.9383	0.9713

**Table 6 sensors-25-01308-t006:** Stability analysis of tower top pressure comparison.

	Mean	Variance	Range
GAN-T2FNN	−0.04237	2.37 × 10^−6^	0.006
LSTM	−0.0439	3.59 × 10^−6^	0.0086
BP neural network	−0.04292	4.25 × 10^−6^	0.009
PID control	−0.04367	6.94 × 10^−6^	0.00964

**Table 7 sensors-25-01308-t007:** Comparative stability analysis of tower top temperature.

	Mean	Variance	Range
GAN-T2FNN	51.146	0.348249	2
LSTM	50.0786	0.675449	3.44
BP neural network	50.193	0.844801	3.79
PID control	50.563	1.21114	4.1

**Table 8 sensors-25-01308-t008:** Comparative stability analysis of precipitated water from atmospheric towers.

	Mean	Variance	Range
GAN-T2FNN	10.79	1.11	3.51
LSTM	11.92	1.38	4.76
BP neural network	12.03	2.16	5.10
PID control	11.42	4.50	8.11

## Data Availability

Data will be made available on request.

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
