# Peer review of "Research on Fuzzy Control of Methanol Distillation Based on SHAP (SHapley Additive exPlanations) Interpretability and Generative Artificial Intelligence"

_sensors, 2025, doi:10.3390/s25051308_

Round 1

Reviewer 1 Report

Comments and Suggestions for Authors

The authors in this paper reported a new way to control the methanol distillation process using SHAP analysis and neural networks. The work is quite interesting and will be useful for chemical manufacturing process. I only have a few concerns before the publication of this work in Sensors:

(1)   PID control is a widely used approach in modern industry. How is the performance of the new control method proposed in this work compared to PID control? If the authors method can outperform PID control than I want to see their comparison in the paper.

(2)   It seems that the optimized temperature or pressure is still not stable according to the performance results shown in Figure 7-8. What is the reason for that?

(3)   If we use authors approach for process control in real industrial manufacturing, which key parameters can we use for tuning? This should be clearly pointed out in the manuscript.

(4)   Please increase the font size of figures. They are too small to see.

Reviewer 2 Report

Comments and Suggestions for Authors

1. In the paper, abbreviations of several phrases are mentioned, and the full name of these abbreviations should not be explained when they first appear, so as not to cause comprehension obstacles to readers in non-professional fields.

2. Some of the pictures in the paper are fuzzy, so it is necessary to adjust them to make them clearly visible.

Author Response

Comments 1: In the paper, abbreviations of several phrases are mentioned, and the full name of these abbreviations should not be explained when they first appear, so as not to cause comprehension obstacles to readers in non-professional fields.

Response 1:

After careful examination of the full text, we have given explanatory notes on the first occurrence of the acronyms SHAP and GMC in the body of the text, on page 3, paragraph 1, and on page 8, paragraph 1, respectively.

Comments 2: Some of the pictures in the paper are fuzzy, so it is necessary to adjust them to make them clearly visible.

Response 2:

We've adjusted the quality of the images in all of our articles to make them clearer and rearranged the layout.

Reviewer 3 Report

Comments and Suggestions for Authors

Gong et al. in this article titled "Research on fuzzy control of methanol distillation based on 2 SHAP interpretability and generative artificial intelligence" proposes a new method to optimize the methanol distillation process, which solves the limitations of traditional control methods (such as PID controllers and BP neural networks) that are difficult to cope with the nonlinear and dynamic characteristics of methanol synthesis by combining type II fuzzy neural networks (T2FNN), generative adversarial networks (GANs) and inverse homeostatic prediction techniques. The content of the manuscript is relatively substantial and the logical structure is also cohesive. It seems to me that after some minor modifications, it is possible to publish on the sensor.

1. In the introduction, the specific shortcomings of traditional control methods (e.g., limited scope of application, slow response to dynamic changes, etc.) can be pointed out more clearly, and the need to use GAN and T2FNN can be highlighted.

2. Although the current experimental results show that GAN-T2FNN is superior to the traditional method, there are few types of comparison models. Comparisons with other advanced control algorithms, such as reinforcement learning control models or other types of neural networks, can be added to enhance the persuasiveness of the experimental conclusions.

3.Although the model performs well in methanol distillation optimization, it may have limitations in other areas, such as scenarios with less data volume or higher real-time requirements. It was suggested that the relevant discussion be supplemented.

4. At present, the innovative description of the combination of T2FNN and GAN is not detailed enough. It is proposed to add a technical description of the points of convergence between the two, as well as an analysis of the key advantages compared to existing methods.

5. In the discussion, the scope of applicability of the method could be further analyzed. For example, how adaptable is the system in a more complex or large-scale industrial environment? Are there other industrial processes that could benefit from a similar approach?

6. If the main innovation of the paper is the GAN-T2FNN control system, it is recommended to set it as a separate core chapter to elaborate on the working mechanism, innovation and implementation process of the method.

7. In the conclusion section, the advantages of the model in terms of MAE, MSE, R², etc., can be re-emphasized, and the potential application scenarios of the method can be supplemented, such as the optimization of other chemical industrial processes.

Reviewer 4 Report

Comments and Suggestions for Authors

You have proposed strategies to solve the problem of traditional control methods in methanol distillation process.

I have some questions as below.

1, in the abstract part, you should clearly show what you will control.

2,in para.4, in the introduction part,  By integrating these three technologies, more stable and accurate 108 temperature control of the methanol reformer can be achieved. The control of the top tem-109 perature and top pressure of the atmospheric column for methanol rectification is ana-110 lyzed by using the interpretability of SHAP, and the influence degree of different param-111 eters combined with the mechanism is clarified.you should also clearly show what parameters that you will control.

3,Table 1,and 2 have the same title.

4,show the details of simulations parameters, structures,environments and the description of the data.

5, analyze the stability of your proposed method by more data.

6,show your innovation deeply.

7,discuss the limitation of your proposed method.

Thank you.

Round 2

Reviewer 4 Report

Comments and Suggestions for Authors

Thank you very much for your modifications!